# Toxin Removal and Inflammatory State Modulation during Online Hemodiafiltration Using Two Different Dialyzers (TRIAD2 Study)

**DOI:** 10.3390/mps4020026

**Published:** 2021-04-22

**Authors:** Gabriele Donati, Maria Cappuccilli, Chiara Donadei, Matteo Righini, Anna Scrivo, Lorenzo Gasperoni, Fulvia Zappulo, Gaetano La Manna

**Affiliations:** Department of Experimental, Diagnostic and Specialty Medicine (DIMES)—Nephrology, Dialysis and Renal Transplant Unit, St. Orsola Hospital IRCCS, University of Bologna, 40138, Bologna, Italy; gabriele.donati@aosp.bo.it (G.D.); maria.cappuccilli@unibo.it (M.C.); chiara.donadei@studio.unibo.it (C.D.); matteo.righini3@studio.unibo.it (M.R.); anna.scrivo@aosp.bo.it (A.S.); lorenzo.gasperoni3@gmail.com (L.G.); fulvia.zappulo@studio.unibo.it (F.Z.)

**Keywords:** cardiovascular mortality, advanced glycation end-products, end-stage renal disease, dialysis membranes, inflammation, online hemodiafiltration, toxins removal

## Abstract

Uremic toxins play a pathological role in atherosclerosis and represent an important risk factor in dialysis patients. Online hemodiafiltration (HDF) has been introduced to improve the clearance of middle- and large-molecular-weight solutes (>500 Da) and has been associated with reduced cardiovascular mortality compared to standard hemodialysis. This non-randomized, open-label observational study will explore the efficacy of two dialyzers currently used for online HDF, a polysulfone-based high-flux membrane, and a cellulose triacetate membrane, in hemodialysis patients with signs of middle-molecule intoxication or intradialytic hypotension. In particular, the two filters will be evaluated for their ability in uremic toxin removal and modulation of inflammatory status. Sixteen subjects in standard chronic bicarbonate hemodialysis requiring a switch to online HDF in view of their clinical status will be enrolled and divided into two treatment arms, according to the previous history of hypersensitivity to polysulfone/polyethersulfone dialysis filters and hypersensitivity to drugs or other allergens. Group A will consist of 16 patients without a previous history of hypersensitivity and will be treated with a polysulfone filter (Helixone FX100), and group B, also consisting of 16 patients, with a previous history of hypersensitivity and will be treated with asymmetric triacetate (ATA; SOLACEA 21-H) dialyzer. Each patient will be followed for a period of 24 months, with monthly assessments of circulating middle-weight toxins and protein-bound toxins, markers of inflammation and oxidative stress, lymphocyte subsets, activated lymphocytes, and monocytes, cell apoptosis, the accumulation of advanced glycation end-products (AGEs), variations in arterial stiffens measured by pulse wave velocity (PWV), and mortality rate. The in vitro effect on endothelial cells of uremic serum collected from patients treated with the two different dialyzers will also be investigated to examine the changes in angiogenesis, cell migration, differentiation, apoptosis and proliferative potential, and gene and protein expression profile. The expected results will be a better awareness of the different effects of polysulfone gold-standard membrane for online HDF and the new ATA membrane on the removal of uremic toxins removal and inflammation due to blood–membrane interaction.

## 1. Introduction

Patients with end-stage renal disease (ESRD) have a significantly increased cardiovascular mortality rate compared to the general population [1,2]. According to the last United States Renal Data System (USRDS) report, in 2018, the prevalence of cardiovascular disease (CVD) in the United States, adjusted for age, sex, and race, was 37.5% of subjects without chronic kidney disease (CKD) vs. 63.4% of patients with stages 1–2 of CKD, 66.6% of patients with stage 3 CKD, and 75.3% of patients with stages 4–5 of CKD [3]. In the natural history of nephropathy, besides the traditional cardiovascular risk factors, an increasingly important role is played by the nontraditional factors, such as renal failure per se, uremic toxin retention, anemia, hyperhomocysteinemia, malnutrition, hyperparathyroidism, electrolyte imbalance, abnormal calcium–phosphate metabolism, chronic inflammation, and oxidative stress [4,5,6,7]. 

Uremic toxins are particularly harmful to the cardiovascular system and represent a relevant risk factor for CVD [8]. Nonetheless, conventional hemodialysis is adequate for the removal of small molecules (<0.5 kDa) such as urea, but not for the middle-molecular-weight solutes in the range of 0.5–60 kDa [9]. A *post hoc* analysis of a major randomized trial, the Membrane Permeabilty Outcome (MPO) study, showed a significant survival benefit associated with the use of high-flux membranes among patients with diabetes or those with albumin levels <4 gr/dL [10]. Online hemodiafiltration (HDF) has been introduced to improve middle-molecule clearance, as larger molecules show a slower removal by diffusion and are more dependent on convective transport. There is some evidence to indicate that HDF with high convection volumes (above 20 L) reduces cardiovascular mortality compared to standard hemodialysis [11,12,13]. However, since middle-molecule clearance through HDF is still inferior compared to a native kidney, the search for novel dialysis technologies is under constant development [14,15]. 

Currently, there is limited evidence behind the identification of middle-molecule intoxication. Many of these products contribute to a homeostatic network in response to other uremia-related changes (e.g., parathyroid hormone) or to further toxins (e.g., the cytokines). Thus, their concentration depends not only on retention but also on endocrine and paracrine corrective mechanisms [16,17]. Vanholder et al. provided comprehensive tables reporting the most important biologic effects and an attributed score per toxin for their evidence of toxicity. The average score for middle-molecular-weight toxins is 2.06 ± 1.19 (maximum = 4). The list is headed by beta-2 microglobulin (B2M), interleukin-6, tumor necrosis factor-alpha, and fibroblast growth factor 23, each of them with a score of four. As to the average evidence score of toxicity, no difference was found between middle molecules and small water-soluble compounds (1.75 ± 1.29), and between middle molecules and protein-bound solutes (2.31 ± 0.95) [18]. Due to their molecular weight, the dialytic removal of middle molecules is only possible using high-flux membranes with relatively large pore size, in either diffusive (hemodialysis) or mixed (convective and diffusive = hemodiafiltration) modes.

High-flux polysulfone membranes have been designed to maximize the clearance of uremic toxins of average molecular weight (>500 Da) with the introduction of changes such as reduced hollow fiber wall thickness (≈35 μm), smaller inner diameter (≈185 μm), and higher pore size (≈3.3 nm) [19]. The chemical composition and the asymmetric structure of the fibers make synthetic membranes preferable for convective therapies, such as high-volume HDF.

Cellulose triacetate (CTA) membranes remain one of the best options for patients with hypersensitivity reactions to synthetic fibers [20]. Although synthetic membranes, especially those based on polysulfone, have a lower risk of hypersensitization than non-biocompatible ones, some cases of adverse reactions have been reported, plausibly due to the contact between the dialyzer itself and plasma proteins and platelets [21]. 

Recently, a new generation filter in asymmetric triacetate (ATA) has been introduced, born from a technology that combines the requirements of synthetic membranes for the possibility of being used with high-volume HDF and the advantages of natural fibers (cellulose) in terms of reduced allergic reactions (Figure 1) [22,23].

These novel ATA membranes have been shown to achieve satisfactory Kt and convection volume, with good biocompatibility and inflammatory profiles, indicating them as valuable options for patients allergic to synthetic membranes under HDF treatment [24].

This study will explore the efficacy of two dialyzers with CE marking currently used for online HDF (polysulfone-based high-flux membrane vs. one ATA membrane) among chronic hemodialysis patients with signs of middle-molecule intoxication or intradialytic hypotension during their current treatment with high-flux hemodialysis. Middle-molecule intoxication will be assessed according to the criteria described by Locatelli et al. [25], focusing on patients with dialysis-related amyloidosis, B2M >30 mg/L, peripheral neuropathy, restless leg syndrome, and major cardiovascular comorbidities requiring an optimal removal of middle molecules [26]. Concerning the risk of intradialytic hypotension, online HDF appears to have a blood-pressure-stabilizing effect as a result of blood cooling via enhanced thermal energy losses within the extracorporeal system, in spite of the use of replacement fluid prepared from pre-warmed dialysate [27,28].

The primary endpoint is the evaluation of the efficacy of the two dialysis filters in terms of the removal of uremic toxins and modulation of inflammatory status using the following parameters:-serum levels of albumin, B2M, C-reactive protein (CRP), myoglobin, light chains, retinol-binding protein, homocysteine, alpha-2 microglobulin (A2M), p-cresol, indoxyl sulfate, bisphenol A (BPA), fibroblast growth factor 23 (FGF23), and inflammatory cytokines;-lymphocyte subsets, activated lymphocytes, activated monocytes, and apoptosis rate;-accumulation of advanced glycation end-products (AGEs) as an index of metabolic and oxidative stress.

Moreover, the study will also evaluate in the patients treated with the two dialyzers: -the prevalence of infectious complications;-the rate of cardiovascular morbidity and mortality;-the rate of hospitalization;-patient survival;-angiogenesis and cell migration ability on in vitro models of cultured endothelial cells exposed to serum from patients treated with the two dialyzers;-the changes in arterial stiffness through pulse wave velocity (PWV).

## 2. Experimental Design and Methods

### 2.1. Study Design

A prospective, observational, open-label, single-center study will be conducted with CE-marked medical devices already used in the clinical practice. The eligible subjects will be 32 patients in standard chronic bicarbonate hemodialysis requiring a switch to online HDF in view of their clinical status—when middle-molecule intoxication or intradialytic hypotension will be diagnosed. This switch will be based only on the clinical status of the patients and not on the study design. Middle-molecule intoxication will be assessed when patients have B2M levels of >30 mg/L, free light chain values of >100 mg/L for κ chains and >50 mg/L for λ chains, peripheral neuropathy, restless leg syndrome, and major cardiovascular comorbidities requiring an optimal removal of middle molecules [25,26,27]. Intradialytic hypotension was defined as (1) systolic blood pressure (SBP) at dialysis starting at ≥100 mmHg and subsequent SBP at <90 mmHg, without symptoms (condition 1); (2) SBP at dialysis starting at ≥100 mm Hg, followed by a decrease of at least 10%, symptomatic (condition 2); and (3) decrease in SBP after dialysis with a start of at least 25 mm Hg, symptomatic, with therapeutic intervention (condition 3) [27].

The duration of participation in the study for each patient enrolled will be 24 months, during which he/she will undergo hemodialysis treatment in HDF three times a week, according to the normal clinical assistance path. Given the non-interventional nature of this study, it must be specified that the prescription of one of the two dialyzers and the allotment of each patient to the treatment group depend exclusively on the individual clinical conditions and the previous history of hypersensitivity to polysulfone/polyethersulfone dialysis filters, and hypersensitivity to drugs or other allergens.

Group A: 16 patients without a previous history of hypersensitivity treated by online HDF with FX100.Group B: 16 patients with a previous history of hypersensitivity treated by online HDF with SOLACEA 21-H.

For each patient, the following demographic, anthropometric, and clinical parameters will be collected at the beginning and monthly throughout the study.-General features and clinical data: sex, age, weight, height, BMI, blood pressure, heart rate, and the Charlson Comorbidity Index.-Previous medical history: renal disease, age at onset, acute or acute on chronic, waiting list for a kidney transplant, and routine laboratory assays.-Physical examination: pulmonary or peripheral hyperhydration, heart rate, and vascular murmurs.-Ongoing medical therapy.

All patients enrolled in the study will be subjected to:Hemodialysis depurative treatment using online HDF technique three times a week according to regular clinical practice;Use of two types of high-flow dialysis filters for online HDF:
FX100 filter (Helixone, Fresenius Medical Care), ultrafiltration coefficient 73 mL/h/mmHg, cut-off 35,000 Da, surface area 2.2 m^2^, steam sterilization;SOLACEA 21-H filter (ATA, Nipro Europe) for polyallergic patients and in particular patients allergic to polysulfone filters, ultrafiltration coefficient 76 mL/h/mmHg, cut-off 45,000 Da, surface 2.1 m^2^, gamma-ray sterilization;Convective volume during dialysis > 20 L per session will be obtained by means of two biofeedback systems, namely Fresenius Medical Care AutoSubPlus™ and Nipro Max-Sub™;Ultrapure dialysis water will be used, defined as <0.1 CFU/mL and <0.03 EU/mL;Heparinization of the extracorporeal circuit, obtained by low-molecular-weight heparin (Inhixa™, enoxaparin sodium, Techdow Pharma, Milan, Italy), at the dosage of 2000 IU in patients with body weight < 50 kg, 4000 IU in patients with weight >50 kg and <90 kg, 6000 IU in patients with weight ≥ 90 kg. Low-molecular-weight heparin will be administered in a single bolus, at the beginning of the dialysis treatment, immediately after the connection of the patient to the extracorporeal circuit;Administration of the appropriate medical therapies, regular laboratory, and instrumental clinical monitoring, as well as possible hospitalization of patients in case of the onset of intercurrent diseases.

For the purposes of the study, each patient will be followed for a period of 24 months after enrolment to collect the following parameters:-monthly laboratory assays, namely circulating medium-molecular-weight uremic toxins and toxins linked to plasma proteins, inflammation markers and cytokines, analysis of lymphocyte subsets, activated lymphocytes and monocytes, and cell apoptosis rate;-monthly assessment of accumulation of AGEs as an index of metabolic and oxidative stress;-infectious complications;-cardiovascular complications;-eventual kidney transplantation;-possible interruption of online HDF and the switch to another dialysis technique;-need for hospitalization; -patient survival;-in vitro effect on endothelial cells of uremic serum collected from patients treated with the two different dialyzers on angiogenesis, cell migration, differentiation, apoptosis and proliferative potential, and gene and protein expression profile;-monthly bioimpedance analysis (BIA);-monthly measurement of changes in arterial stiffness by PWV.

### 2.2. Patients 

The study population will consist of ESRD patients who need to switch from standard bicarbonate dialysis to online HDF with a thrice-weekly rhythm. Sixteen eligible patients will be enrolled and divided into two treatment groups (filter FX100 vs. filter SOLACEA 21-H), according to the individual clinical conditions. The patients will be enrolled consecutively in order to reach 16 subjects per group. 

In case of drop out, another patient will be recruited to maintain the total number of 32 patients to be evaluated over 24 months.

The inclusion criteria are as follows: (i) ESRD patients under thrice-weekly standard bicarbonate hemodialysis; (ii) residual diuresis < 200 mL/day; (iii) age > 18 years; (iv) vascular access for hemodialysis (FAV) with blood flow ≥ 300 mL/min; (v) need for treatment with HDF online for signs of toxicity from middle molecules (B2M > 30 mg/L, peripheral neuropathy, cardiovascular comorbidity) or intradialytic hypotension; and (vi) written informed consent to participate in the study.

Patients with acute coronary syndrome, acute hemorrhage, recent surgery, active infections, malignancy, or other pathologies with unfavorable short-term prognosis—or who are enrolled in other studies—will not be considered eligible. Patients undergoing dialysis by means of a tunneled cuffed catheter or arteriovenous graft will not be considered eligible as well, since these kinds of vascular accesses per se can elicit inflammation [7]. In case of complications of the arteriovenous fistula (i.e., fistula stenosis) recovered by angioplasty without a temporary or tunneled catheter, the patient can continue to participate in the study. When a temporary catheter for dialysis will be used for a period >1 week, that period will not be considered for the study, and blood will not be drawn for the laboratory findings until the arteriovenous fistula has recovered its function. In case of the dropout of a participant due to a clinical event, another patient will be enrolled to maintain the same sample size for each group.

The time in which each patient will be engaged during the dialysis day scheduled by the study coincides with the time of the dialysis session that will begin at 8 a.m. until 12 a.m. 

There are no contraindications to concomitant therapies that will be administered according to normal clinical practice and registered in the management database.

Each patient is fully entitled to terminate his or her participation in the study at any time. 

The schedule of examinations and evaluations over the 24 months of patient participation in the study are shown in Table 1.

### 2.3. Specimen Collection

As detailed in Figure 2, blood samples will be taken monthly from each patient before the dialysis session for the routine laboratory assays. For the study-specific assessments, an additional 20 mL of blood will be drawn before and after the dialysis session at T0, T1, T3, T6, T12, and T24 (last treatment). In addition, to check the hypersensitivity-induced complement activation and the related transient neutropenia occurring in the first 30 min of hemodialysis [29], neutrophil and eosinophil counts will be assessed 15 min after the start of the dialysis session. In order to comply with the observational nature of this study, blood specimens will be collected from the fistula needle at the time of the scheduled dialysis sessions, with no need of additional venipuncture. Following the isolation of serum, plasma, and PBMC (peripheral blood mononuclear cells), samples will be bio-banked at −80 °C until analysis. 

### 2.4. Laboratory Assays

The determination of the serum levels of albumin, B2M, CRP, myoglobin, light chains, retinol-binding protein, and homocysteine will be carried out according to the standard procedures. 

P-cresol, indoxyl sulfate, and BPA will be assayed using the liquid chromatography-mass spectrometry (LC-MS) method at the Metropolitan unified laboratory (LUM), as previously described [30,31,32].

The circulating levels of FGF23, A1M, and inflammatory cytokines IL-1 β, IL-6, IL-10, IL-12 (p70), IL-17, TNF-α, and IFN-γ will be measured using Luminex MAGPIX^®^ technology (Millipore Corp, Billerica, MA, USA) according to the manufacturer’s indications. 

Flow cytometry (CytoFLEX S, Beckman Coulter, Milan, Italy) will be used to analyze:-lymphocyte subsets through BD Simultest™ IMK-Lymphocyte, a two-color direct immunofluorescence reagent kit for enumerating percentages of the following cell types: T (CD3+) lymphocytes, B (CD19+) lymphocytes, helper/inducer T (CD3+CD4+) lymphocytes, suppressor/cytotoxic T (CD3+CD8+) lymphocytes, natural killer (NK) (CD3-CD16+ and/or CD56+) lymphocytes, and CD3+CD4+/CD3+CD8+ ratio;-activated lymphocytes CD3+HLA-DR+ and CD8+CD57+;-the expression on monocytes of the activation markers CD11b, HLA-DR, and CD14 [7], as well as senescence marker CD32 [33];-the detection of apoptosis through the classical Annexin V method [34].

### 2.5. Instrumental Examinations

#### 2.5.1. Bioimpedance Analysis 

BIA will be performed monthly to non-invasively measure extracellular, intracellular, and total body water volumes before and after dialysis sessions, using the Akern electrofluid graph (EFG) device (Akern SRL., Pontassieve, Firenze, Italy).

#### 2.5.2. Measurement of AGE Accumulation through Skin Autofluorescence

All patients will also be assessed monthly for AGE accumulation by measuring cutaneous autofluorescence test through an AGE Reader (AGE Reader, Diagnoptics Technologies B.V., Groningen, the Netherlands). The test is rapid, non-invasive, accurate, and repeatable, and is performed in less than 30 s by placing the patient’s forearm on the measurement window, where a skin surface of about 4 cm^2^ is illuminated by a UV light source with excitation wavelengths of 300–420 nm (peak length: 350 nm). The measurement will be carried out in the non-fistula arm at room temperature. The mean of three consecutive measurements will be computed.

#### 2.5.3. Measurement of Arterial Stiffness through Pulse Wave Velocity

PWV reflects the velocity of the systolic wave through the arteries, and it will be used to stratify the risk of cardiovascular events in the two groups of patients. The instrument (PWV, Complior SP, Artech Medical, Pantin, France) operates through a system composed of two sensors: the first is placed on the carotid pulse and the second is placed on the femoral or on the radial pulse. Systolic wave velocity is measured: (a) from carotid to femoral or (b) from carotid to radial. Normally, the first measure is the most used. The procedure lasts around 10 min [35,36].

### 2.6. In Vitro Model

The in vitro model of human coronary artery endothelial cells (HCAECs) will be developed to investigate the effects of the exposure to serum from dialysis patients treated with the two different dialyzers in the induction of angiogenesis, cell migration, differentiation, apoptosis, proliferative potential, and gene and protein expression. HCAECs are the ideal candidates for the study of endothelial cell functions and metabolism. The cells will be cultured in a growth medium enriched with 10% of serum specimens collected from patients before and after dialysis session at times T0, T12, and T24.

To investigate the effect of sera collected from patients treated with Helixone or ATA filters, the following parameters will be evaluated:Cell proliferation determined by bromodeoxyuridine (BrdU) integration assay. It allows the visualization of the proliferating cells in relation to the total number of cells through detection by immunofluorescence;Migration of endothelial cells using the modified Boyden chamber method. Endothelial cells move through a process called chemotaxis, along with a gradient of factors that induce angiogenesis. In this assay, the cells are plated on a porous surface that separates two compartments across which cells can migrate in response to an angiogenic factor placed in the chamber below;The ability of endothelial cells to self-organize into capillary-like structures. Cells will be grown on Matrigel matrices to assess adhesion, migration, and tubule organization. In addition, three-dimensional (3-D) matrices will also be used to mimic the in vivo angiogenic system. Initially, the endothelial cells form tubules in the horizontal plane and after 12 or more days, the endothelial tubules begin to ramify upward and penetrate the gel to form a 3-D tubule network;Gene expression of vascular endothelial growth factor (VEGF), vascular endothelial growth factor receptor-1 (VEGFR1), vascular endothelial growth factor receptor-2 (VEGFR2), and the changes occurring in the different experimental conditions by real-time PCR using TaqMan Probes on an iCycler iQ Real-Time PCR Detection System (Bio-Rad, Hercules, CA, USA). VEGF and its receptors constitute the major regulatory system of vascular development and play a central role in vasculogenesis, as well as in both physiological and pathological angiogenesis. This system is expressed in many tissues and cells when angiogenesis is in progress, while it is restrained when the angiogenic process is inactive;Protein expression of VEGF and its receptors in endothelial cells and the variations they undergo under different experimental conditions evaluated by means of immunocytochemistry;Angiogenic growth factors, such as fibroblast growth factor 2 (FGF2), VEGF, Thrombospondin-1 (TSP-1), Troponin I, IFN-α, IFN-γ, IL-4, IL-12, and retinoic acid in the supernatant of cell cultures through Luminex MAGPIX^®^ technology (Millipore Corp), which allows both monoplex and multiplex assays of proteins and nucleic acids.

### 2.7. Statistical Analysis

Clinical and biological data will be recorded in a dedicated database for successive statistical analysis. Continuous variables will be expressed as mean and standard deviation if normally distributed, or as a median with minimum–maximum range if unusually distributed. Categorical variables will be presented as absolute numbers and percentages. The Kolmogorov–Smirnov test will be used to test the normality of the sample distribution.

Continuous variables will be computed using parametric (Student’s *t*-test, ANOVA) or non-parametric (Wilcoxon rank-sum test, Mann–Whitney U test) tests, and categorical variables will be computed through chi-square.Statistical analysis will be performed by Prism software (version 8 for Apple, GraphPad Software Inc., La Jolla, CA, USA), with a significance level set at *p* < 0.05.

### 2.8. Rules of Good Clinical Practice

This study will be conducted according to the principles of Good Clinical Practice (GCP; ICH Harmonized Tripartite Guidelines for Good Clinical Practice 1996 Directive 91/507/EEC; DM 15.7.1997), the Helsinki Declaration, and national regulations on the conduct of clinical trials. By signing the protocol, the investigator agrees to adhere to the procedures and instructions contained therein and to carry out the study according to GCP, the Helsinki Declaration, and national regulations governing clinical trials.

### 2.9. Ethics and Dissemination

The study protocol was approved by the Area-Vasta Emilia-Romagna Centro Ethics Committee with approval number 596/2019/Oss/AOUBo on 11 September 2019. Written informed consent will be obtained from all participants. The researchers will identify the subject with a code, and the data collected during the study will be recorded, processed, and stored for seven years after the conclusion of the trial together with this code. 

The results will be disseminated only in a strictly anonymous or aggregated form through publications in peer-reviewed journals and scientific congresses.

## 3. Discussion

This non-randomized, open-label observational study will compare two types of dialyzers currently in use for online HDF, FX100 (Helixone™, Fresenius Medical Care) vs. SOLACEA 21-H (ATA™, Nipro Europe), for their efficacy in terms of toxin removal and control of inflammation. In particular, the following parameters will be investigated: (a) removal of middle-weight uremic toxins; (b) retention of BPA; (c) AGE levels; (d) inflammatory cytokine profile; (e) activation of lymphocytes and monocytes and apoptosis induction; and (f) morbidity and mortality rate. Notably, the comparative evaluations will be carried out per single session with pre- vs. post-dialysis assessments during the 24 month follow-up.

The participants will be recruited among chronic hemodialysis patients who usually undergo high-flux hemodialysis with clinical laboratory signs of middle-molecule intoxication or intradialytic hypotension 

The main clinical application of online HDF is the removal of uremic toxins in the range of the middle molecule (500–60,000 Da), with the purpose of increasing the survival of chronic uremic patients and reducing chronic inflammation and the rate of accelerated atherosclerosis, while also considering the comorbidity burden of dialysis, namely old age (>75 years), diabetes, and cardiovascular disease [18]. Among the middle-weight uremic toxins, B2M is historically the most studied, and it has been correlated with increased overall mortality when its serum levels are above 27.5 mg/L [37,38]. Moreover, free light chains, also classified as middle molecules, have been proven to trigger impairment of neutrophil function, contribute to the chronic inflammation state of uremic patients, and increase the risk of bacterial infections or vascular calcifications [39,40,41]. High circulating levels of FGF23 have been also correlated with cardiovascular mortality in CKD and dialysis patients [42,43,44]. 

In this scenario, a central point is represented by the choice of the ideal dialyzer for online HDF, as the use of synthetic polysulfone membranes can increase the risk of hypersensitivity reactions in up to 4.2 out of 1000 HDF sessions [45]. In the last few years, growing attention has been afforded to another component of the dialysis polysulfone membrane: bisphenol A (BPA, molecular weight 228.3 kDa protein-bound molecule) [46,47]. BPA is considered an endocrine-disrupting chemical that acts mainly as a female hormone: it is associated with reproductive dysfunction, immune abnormality, and increased incidence of cancer. It is also a ubiquitous environmental toxicant found in plastic food and beverage containers. Patients on hemodialysis are a high-risk population for BPA overload for two main reasons. First, the normal urinary elimination of the conjugated BPA is abolished [48]. Second, BPA is leached by various types of polysulfone-derived dialyzers from both the membrane and the housing components [49]. 

Recently, a new generation filter in asymmetric triacetate (ATA) has been introduced, born from a technology that combines the requirements of synthetic membranes for the possibility of being used with high-volume HDF and the advantages of natural and BPA-free fibers (cellulose) in terms of reduced allergic reactions [22,23,24]. 

In order to compare the effects of the two dialysis filters on inflammation, the study will also focus on the accumulation of AGEs as an index of carbonyl and oxidative stress. The activation and up-regulation of the AGE receptors, including the Receptor for AGE (RAGE), leads to the initiation of inflammatory cascades involved in the formation of atherosclerotic plaques and endothelial dysfunction and in the pathogenesis of cardiovascular disease. AGEs are also related to the thickening of the intima–media tunica of the carotid arteries, considered a marker of arteriosclerosis, as clinically confirmed by the close relationship between coronary artery disease and atherosclerotic complications in diabetes and renal failure [50,51,52]. Patients undergoing chronic dialysis treatment seem to be more prone to chemical modifications of proteins and the accumulation of AGEs [53,54]. AGE levels are an independent predictor of overall and cardiovascular mortality as pre-existing cardiovascular disease, C-reactive protein, and serum albumin during a three-year follow-up in 109 HDF patients [55].

The blood–membrane contact during dialysis triggers a foreign body reaction with the recruitment of neutrophils and monocytes, which release pro-inflammatory cytokines; therefore, the measurement of neutrophils and monocytes and of their activation or apoptosis provides useful information on dialyzer biocompatibility [32,56,57]. High-flux polysulfone membranes are especially implicated in the rise in circulating inflammatory indexes, namely C-reactive protein (CRP), IL-1β, TNF-α, IL-6, and carbonylation rate in dialysis patients [58,59,60]. The biocompatibility and flow properties of the different filters can influence the inflammatory and oxidative response to dialysis and the levels of oxidized molecules [61,62]. A study carried out by Ojeda et al. compared the biocompatibility of four membranes (polyamide, polynephron, helixone, and CTA) in online HDF for four weeks. Biocompatibility was similar among the membranes, showing no sustained differences in complement, platelet, monocytes, CRT, or plasma levels of IL-1, IL-6, IL-10, PAI-1, ICAM1, or VCAM1 [63]. Nonetheless, it is not possible to point out the effect of membrane biocompatibility in the long term. To explore possible consequences of uremic toxin retention and inflammation-accelerated atherosclerosis, a secondary objective of the study is the analysis of angiogenesis and cell migration on models of cultured endothelial cells exposed to sera from patients treated with the two dialyzers, as well as PWV measurement in the 24 months follow-up [64]. 

PWV is the most common way to measure arterial stiffness, as it determines pressure wave propagation velocity from the aorta toward peripheral arterial branches. In particular, PWV is a stronger predictor of cardiovascular events and all-cause mortality than ambulatory blood pressure in 170 chronic hemodialysis patients enrolled during a mean follow-up of 28 ± 11 months [36]. In a cohort of 241 hemodialysis patients, PWV values > 12.0 m/s were independently associated with all-cause mortality (Hazard Ratio 5.4; 95% CI 2.4–11.9) and cardiovascular mortality (Hazard Ratio 5.9, 95% CI 2.3–15.5) during a mean follow-up of 72 months [35]. No study at present is available to point out the role of the kind of dialyzer on pulse wave velocity values in a long-term follow-up.

## 4. Conclusions 

To evaluate the potential impact of the trial, we must consider that all previous studies indicate online HDF as the best option to reduce cardiovascular mortality in hemodialysis patients. This improved survival likely depends on the more effective removal of uremic toxins in the range of the middle molecules and in lower activation of chronic inflammation in comparison to standard hemodialysis. The aim of this study is to explore in depth the effects on toxin removal and inflammation of two types of dialyzers that differ for the chemical composition and the polysulfone gold-standard membrane for online HDF and the new ATA membranes. Despite its novelty, this study has some limitations, in particular the observational nature of the design and the limited sample size, mainly due to the relatively low number of allergic patients in our center. Further studies on a larger population are needed to assess the effects of these two dialysis membranes over a longer term and to provide a patient-tailored approach to subjects with a previous clinical history of hypersensitivity to synthetic filters, drugs, or other allergens.

## Figures and Tables

**Figure 1 mps-04-00026-f001:**
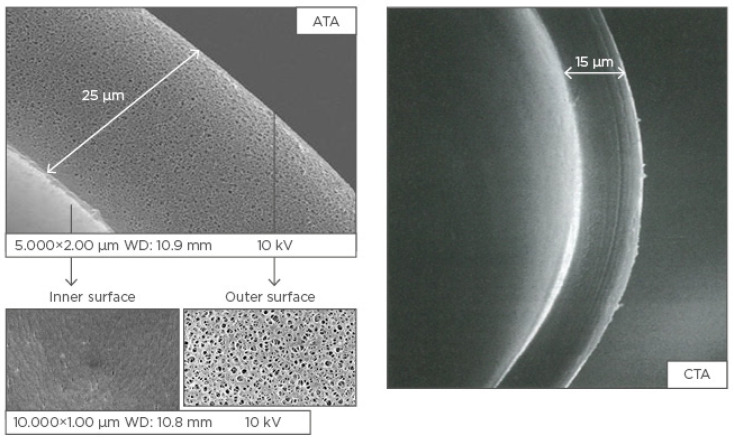
Structures of the asymmetric triacetate (ATA) and cellulose triacetate (CTA) membrane. Adapted with permission from Ref. [23]. Copyright 2020 S. Karger AG.

**Figure 2 mps-04-00026-f002:**
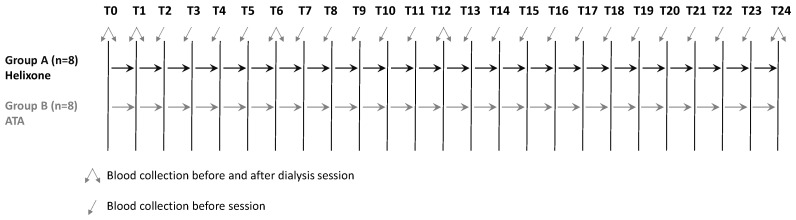
Timetable of specimen collection.

**Table 1 mps-04-00026-t001:** Timetable of visits and evaluations.

Visit Timing	T0	T1	T2	T3	T4	T5	T6	T7	T8	T9	T10	T11	T12	T13	T14	T15	T16	T17	T18	T19	T20	T21	T22	T23	T24
Enrollment	X																								
Informed consent	X																								
Clinical history	X	X	X	X	X	X	X	X	X	X	X	X	X	X	X	X	X	X	X	X	X	X	X	X	X
Therapies	X	X	X	X	X	X	X	X	X	X	X	X	X	X	X	X	X	X	X	X	X	X	X	X	X
General assessment	X	X	X	X	X	X	X	X	X	X	X	X	X	X	X	X	X	X	X	X	X	X	X	X	X
Lab tests	X	X		X			X						X												X
AGEs measurements	X	X	X	X	X	X	X	X	X	X	X	X	X	X	X	X	X	X	X	X	X	X	X	X	X
BIA	X	X	X	X	X	X	X	X	X	X	X	X	X	X	X	X	X	X	X	X	X	X	X	X	X
PWV	X	X	X	X	X	X	X	X	X	X	X	X	X	X	X	X	X	X	X	X	X	X	X	X	X
Adverse events	X	X	X	X	X	X	X	X	X	X	X	X	X	X	X	X	X	X	X	X	X	X	X	X	X

Abbreviations: AGEs, Advanced glycation end-products; BIA, Bioimpedance analysis; PWV, pulse wave velocity; T0, baseline; T1, month 1; T2, month 2; T3, month 3; T4, month 4; T5, month 5; T6, month 6; T7, month 7; T8, month 9; T10, month 10; T11, month 11; T12, month 12; T13, month 13; T14, month 14; T15, month 15; T16, month 16; T17, month 17; T18, month 18; T19, month 19; T20, month 20; T21, month 21; T22, month 22; T23, month 23; T24, month 24 (end of the study).

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
