# Peer review of "Toxin Removal and Inflammatory State Modulation during Online Hemodiafiltration Using Two Different Dialyzers (TRIAD2 Study)"

_mps, 2021, doi:10.3390/mps4020026_

Round 1
Reviewer 1 Report
In this paper Donati et al present the protocol for an observational study about uremic toxin clearance and systemic inflammation in patients switching from conventional high flux (HF) hemodialysis (HD) to online hemodiafiltration (HDF).
16 subjects will be split in 2 groups (n=8): patients allergic to polysulfone (PS) will use asymmetric triacetate (ATA) and will be compared to patients undergoing HDF with PS filters.
The study design present several limitations:
- the number of patient is particularly low. The authors do not perform any power analysis. While it is unlikely that with n=8 any inflammatory parameter will be significantly changed, it is almost impossible that any secondary endpoint (e.g mortality or CV events) will be significantly different between the 2 groups
- the study does not seem observational in nature. Switching from HD to HDF is a therapeutic intervention as it is the choice of the dialysis membranes
- the level of evidence behind the identification of "middle molecule intoxication" is low. This should be reported. Moreover, it underlines once again the experimental nature of the protocol
- the vast number of variables investigated in a such small patient group may induce the identification of random differences . Statistical corrections should be implemented to avoid them
- the study design does not include any matching. This will make more difficult any form of comparison between the 2 groups
- arterio-venous fistula (AVF, not FAV) function cutoff of 250cc/min seems way to small as it is below the standard performance of a dialysis catheter
- the discussion is particularly long and seems a review about the chosen filters. For instance the BPA paragraph is not relevant to the protocol as the authors will not measure BPA. Thus should be synthesized .
Author Response
Dear reviewer,
The reply to your concerns is provided in the attached file.
Thanks for your comments that allowed us to subsustantially improve the manuscript. We hope that all the points have been addressed properly.
Kind regards.

Reviewer 2 Report
The protocol by Donati et al is an interesting, ambitious, and extensive proposal for a clinical trial. While the research does appear to have some merits, there are multiple things that should be elucidated more clearly in the text. In addition, it appears the chosen experimental design does not support the stated goal of the study. However, considering that the protocol was already approved by MREC at the end of 2019, it seems that input from this peer review cannot be addressed in the original clinical research.
Major comments
- At the end of the abstract, it is stated that “The main goal is optimization of the choice of the most appropriate dialyzer to the individual previous history and clinical condition of each subject using a patient-tailored approach”. It is really difficult to draw any conclusions from the current setup as patients are not allocated randomly and thus there is indication bias. In addition, the number of patients is low. I doubt whether this low number is sufficient to detect any difference. Please motivate the number of patients (e.g. sample size and power calculation)?
- Who assesses the need to switch to HDF (therewith eligibility for enrollment)? Their own dialysis doctor? An independent expert? The researcher?
Minor comments
- Please explain more clearly why you also focus on protein bound toxins. Literature is inconsistent about the question whether HDF improves protein-bound toxin clearance.
E.g. the sentence in the introduction: The increasing attention given to techniques able to improve middle molecule clearance is due to the huge pattern of protein-bound toxins and middle molecular weight uremic toxins, known to contribute to uremic syndrome in terms of cardiovascular complications, inflammation and fibrosis [18]. Accumulation of protein-bound toxins in dialysis patients(contributing to the uremic syndrome) can not be an explanation for the increasing attention for improved middle molecule clearance.
And in the discussion: Finally, the accumulation of indoxyl sulphate and p-cresyl sulphate, the most important representatives of the protein-bound toxins, is predictive of cardiovascular and all-cause mortality in uremic patients [41, 42. What is the purpose of this sentence in the context of this study?
- Remark about the following sentence in the introduction: The increasing attention given to techniques able to improve middle molecule clearance is due to the huge pattern of protein-bound toxins and middle molecular weight uremic toxins, known to contribute to uremic syndrome in terms of cardiovascular complications, inflammation and fibrosis [18]. Please replace ‘pattern’ by ‘amount’.
- Introduction --> Sentence “A post hoc analysis … with albumin levels <4 gr/dL.” is not connected to the rest of the text. The relationship to the other statements in this paragraph should be made clear to give this sentence meaning and relevance.
- With regard to the patient population: please explain the rationale for selecting patients with intradialytic hypotension already in your introduction (not only in your discussion).
- There are various things that should be somewhat more elucidated in the methods section
- Please define ‘signs of middle molecule intoxication’ and ‘intradialytic hypotension’ already in your methods section (not (only) in your discussion section).
- Inclusion criterium is shunt flow >250mL/min. What happens if flow decreases during the study period? Intervention? How long do you tolerate shunt flow <250 mL before exclusion should this occur in a patient during study period?
- It is unclear why patients with an arteriovenous graft are excluded.
- AGE measurements --> Are these performed on a specific patch/site of skin for all patients? Or at least the same patch/site for a single patient? Is it measured at multiple sites each time or just a single spot?
- In vitro model --> the remark that HCAECs are ideal candidates for the study of endothelial cell function and metabolism should be supported by at least one reference and/or an explanation by the authors. Moreover, the in vitro model will be developed? So this is actually not a pre-existing (validated) test? Is it wise to use an unvalidated test then for such a trial? Are there validated alternatives? If so, why not choose these test(s)? Why only use 10% of patient serum in your cell assays, why not more to increase the chance of detecting a difference between groups?
- The discussion is at times repetitive and/or unclear with various remarks seemingly out of the blue with no coherence to the surrounding text. I would like to advise the authors to critically revise the discussion section to address this. Examples are lines 179-180 (why is this relevant? What is the relevance? Should be explained), lines 181-184 (difficult to read sentence, not connected to the surrounding text, essentially a one-sentence paragraph), lines 197-200 (unclear what the authors mean here, seems to be missing a/some word(s)), and lines 236-239 (stronger predictor compared to… what exactly?).
- The study compares toxin removal between 2 dialyzers. Please discuss literature with regard to differences in toxin removal between these 2 dialyzers (or state that information in literature with regard to this subject is lacking).
Author Response

(The authors gave the same response as above.)

Round 2
Reviewer 1 Report
The manuscript has been improved by authors' edits. Some of the criticisms persist but are intrinsic to the type of paper
Author Response
The manuscript has been improved by authors' edits. Some of the criticisms persist but are intrinsic to the type of paper.
Thanks for your understanding. We are aware that the trial has some weak points, mainly due to the limited sample size and its observational design. We added paragraph at the end of the conclusions section about the limitations of the study and the future perspectives.
Reviewer 2 Report
Most of my comments have been adequately addressed. I have a few minor comments.
Several sentences are incorrect and/or difficult to understand and have to be rewritten:
- Line 84-86: ‘ although – the cytokines’
- Line 95-98: ‘ Due – mode’: I would remove change it into ‘Due to their molecular weight, the dialytic removal of middle molecules is only possible using high flux membranes with a relatively large pore size, in either diffusive (hemodialysis) or mixed (convective and diffusive = hemodiafiltration) mode.’
- Line 111-113: The features of synthetic membranes related their chemical composition, and the asymmetric structure of the fibers make them preferable for new highly-convective therapies, such as high-volume HDF
- Line 115: remove ‘ however’
- Line 181-183: Intradialytic hypotension 181 was defined as (1) systolic blood pressure (SBP) at dialysis start ≥100 mm Hg and subsequent SBP ≥90 mm Hg, without symptoms (condition 1) --> what do you mean; where is the hypotension? Do you mean ‘subsequent SBP <90 mmHg without symptoms’, or ‘subsequent SPB ≥90 mm Hg and <100 mmHg’?
- Line 284-286: Another one to maintain the same number of patients for each group will replace a patient if he/she should leave the study for a clinical event.
The discussion is much better now. Two small remarks:
- Please mention that the new ATA membrane is BPA-free in the paragraph about the ATA membrane (line 507-510)
- Please discuss limitations of your study
Suggestion for the authors (feel free to adapt it or not): also check leukocyte count after 15-30 min in some of dialysis sessions to check for type B dialyzer reactions which are related to complement activation and typically lead to transient neutropenia with a nadir in the white cell count after ~15 min (see N Engl J Med. 1984;311(14):878. PMID 6332276).
Author Response
Several sentences are incorrect and/or difficult to understand and have to be rewritten:
Line 84-86: ‘ although – the cytokines’
Thanks to your suggestion, the sentence has been changed in order to make it clearer.
Line 95-98: ‘ Due – mode’: I would remove change it into ‘Due to their molecular weight, the dialytic removal of middle molecules is only possible using high flux membranes with a relatively large pore size, in either diffusive (hemodialysis) or mixed (convective and diffusive = hemodiafiltration) mode.’
The sentence has been rewritten in a clearer way.
Line 111-113: The features of synthetic membranes related their chemical composition, and the asymmetric structure of the fibers make them preferable for new highly-convective therapies, such as high-volume HDF
The sentence has been simplified.
Line 115: remove ‘ however’
Done.
Line 181-183: Intradialytic hypotension 181 was defined as (1) systolic blood pressure (SBP) at dialysis start ≥100 mm Hg and subsequent SBP ≥90 mm Hg, without symptoms (condition 1) --> what do you mean; where is the hypotension? Do you mean ‘subsequent SBP <90 mmHg without symptoms’, or ‘subsequent SPB ≥90 mm Hg and <100 mmHg’?
This was a typing mistake, we meant SBP at dialysis start above 100 mmHg and then SBP below 90 mmHg, without symptoms.
Line 284-286: Another one to maintain the same number of patients for each group will replace a patient if he/she should leave the study for a clinical event.
Thanks to your suggestion, the sentence has been changed in order to make it clearer.
The discussion is much better now. Two small remarks:
Please mention that the new ATA membrane is BPA-free in the paragraph about the ATA membrane (line 507-510)
Done.
Please discuss limitations of your study.
We have added two sentences at the end of the conclusions section to clarify the limitations of the study and the future perspectives.
Suggestion for the authors (feel free to adapt it or not): also check leukocyte count after 15-30 min in some of dialysis sessions to check for type B dialyzer reactions which are related to complement activation and typically lead to transient neutropenia with a nadir in the white cell count after ~15 min (see N Engl J Med. 1984;311(14):878. PMID 6332276).
We welcome and agree with this interesting suggestion and added this additional analysis (page 9, lines 295-297). An additional test does not compromise the observational nature of the trial as all the blood will be taken from the fistula needle at the time of the scheduled dialysis sessions without any necessity of a study-specific venipuncture.